# Hypoxia Exacerbates Inflammatory Signaling in Human Coronavirus OC43-Infected Lung Epithelial Cells

**DOI:** 10.3390/biom15081144

**Published:** 2025-08-08

**Authors:** Jarod Zvartau-Hind, Hassan Sadozai, Hateem Z. Kayani, Animesh Acharjee, Rory Williams, Phillip Gould, Christopher A. Reynolds, Bernard Burke

**Affiliations:** 1Research Centre for Health & Life Sciences, University of Coventry, Coventry CV1 2DS, UK; hassan.a.sadozai@gmail.com (H.S.); hateem1989@yahoo.com (H.Z.K.); william31@uni.coventry.ac.uk (R.W.); ac2921@coventry.ac.uk (P.G.); ad5291@coventry.ac.uk (C.A.R.); 2Cancer and Genomic Sciences, School of Medical Sciences, College of Medicine and Health, University of Birmingham, Birmingham B15 2TT, UK; a.acharjee@bham.ac.uk

**Keywords:** COVID-19, hypoxia, inflammation, cytokine storm, HCoV-OC43, CCL20, VEGF, IGFBP3

## Abstract

Cytokine storm (CS) is associated with poor prognosis in COVID-19 patients. Hypoxic signaling has been proposed to influence proinflammatory pathways and to be involved in the development of CS. Here, for the first time, the role of hypoxia in coronavirus-mediated inflammation has been investigated, using transcriptomic and proteomic approaches. Analysis of the transcriptome of A549 lung epithelial cells using RNA sequencing revealed 191 mRNAs which were synergistically upregulated and 43 mRNAs which were synergistically downregulated by the combination of human *Bet**acoronavirus* OC43 (HCoV-OC43) infection and hypoxia. Synergistically upregulated mRNAs were strongly associated with inflammatory pathway activation. Analysis of the expression of 105 cytokines and immune-related proteins using antibody arrays identified five proteins (IGFBP-3, VEGF, CCL20, CD30, and myeloperoxidase) which were markedly upregulated in HCoV-OC43 infection in hypoxia compared to HCoV-OC43 infection in normal oxygen conditions. Our findings show that COVID-19 patients with lung hypoxia may face increased risk of inflammatory complications. Two of the proteins we have identified as synergistically upregulated, the cytokines VEGF and CCL20, represent potential future therapeutic targets. These could be targeted directly or, based on the novel findings described here by inhibiting hypoxia signaling pathways, to reduce excessive inflammatory cytokine responses in patients with severe infections.

## 1. Introduction

The effect of hypoxia (low oxygen) on mRNA and protein expression changes in lung epithelial cells infected with the human coronavirus HCoV-OC43 has not previously been investigated in detail. Here we demonstrate that hypoxia has profound and wide-ranging effects, particularly on inflammatory mediators linked to the serious pathological condition cytokine storm (CS).

Serious cases of COVID-19, caused by infection with severe acute respiratory syndrome coronavirus (SARS-CoV-2), are associated with CS [1]. First used to describe the effects of graft-versus-host disease, CS is a hyperactive, run-away inflammatory response characterized by the excessive release of inflammatory mediators [2,3]. Although the incidence of CS is likely under-reported, a study conducted early in the COVID-19 pandemic (March–September 2020) found that 14.3% of hospitalized COVID-19 patients experienced CS [4]. CS has been observed in patients with Middle East respiratory syndrome-related coronavirus (MERS-CoV), severe acute respiratory syndrome coronavirus (SARS-CoV) and influenza virus infections [5], as well as conditions such as sepsis [6]. Clinical studies examining serum cytokine levels have identified COVID-19 hyperinflammation as being characterized by significantly elevated concentrations of a range of cytokines, including G-CSF, HGF, IFN-α2, IFN-γ, IL-10, IL-12, IL-17, IL-1ra, IL-1α, IL-1β, IL-2, IL-2R, IL-4, IL-6, IL-7, IL-8, IP-10, MCP-1, M-CSF, MIP-1α, MIP-3α (herein referred to as CCL20), MIP-3β, PDGF-BB, and TNF [7,8,9,10,11]. Elevated levels of many of these cytokines have also been identified as predictors of mortality in severe COVID-19 cases [7,8].

Damage to lung tissue caused by SARS-CoV-2 replication and persistent excess immune cell infiltration contributes to the development of acute respiratory distress syndrome (ARDS), a condition characterized by lung epithelial and endothelial damage, inflammation, apoptosis, necrosis and increased alveolar-capillary permeability, leading to alveolar edema [12]. In the early phase of the COVID-19 pandemic, 75% of patients who were transferred to the intensive care unit (ICU) had ARDS and 90% of patients who died in the ICU had ARDS [13].

ARDS is diagnosed using the Berlin 2012 ARDS diagnostic criteria, with one of the major diagnostic hallmarks being the degree of hypoxemic respiratory failure [14]. When oxygen falls below normal levels within a cell, hypoxia-related signaling occurs which allows the cell to adapt to an oxygen-scarce environment. Hypoxia has been shown to regulate the expression of hundreds of genes [15]. This signaling is principally facilitated by a group of transcription factors called hypoxia-inducible factors (HIFs) [16].

Hypoxic signaling has been proposed to influence the inflammatory process via interaction with proinflammatory pathways such as the NF-κB signaling pathway and has been shown to exacerbate lung inflammation [17,18,19] including in the development of CS in patients with COVID-19 [20]. Hypoxia is also capable of synergizing with other stimuli to markedly enhance inflammatory responses [21,22]. There is evidence that this may be due in part to the ability of some of these stimuli to directly modulate HIFs [23,24]. These findings suggest that COVID-19 patients with lung hypoxia may face a greater risk of inflammatory complications [25].

Hypoxia has been identified as a risk factor in various chronic and acute airway diseases. For example, chronic intermittent hypoxia, such as that observed in obstructive sleep apnea, has been shown to exacerbate airway hyperresponsiveness, elevate the expression of proinflammatory cytokines, and promote immune cell infiltration in in vivo models of asthma [26,27]. Moreover, infection of epithelial cells with an Influenza H1N1 virus under hypoxic conditions leads to increased expression of the proinflammatory cytokines TNF and IL-6, and a decrease in the anti-inflammatory cytokine IL-10 [23].

CS and ARDS following COVID-19 infection create a harmful cycle of epithelial injury and deteriorating gaseous exchange in the lungs, exacerbating hypoxia [28], which may then further exacerbate this damaging cycle of hyperinflammation. Identifying inflammatory markers upregulated by hypoxia may be of utility in the development of future therapies to ameliorate CS in SARS-CoV-2 and other infections, as an addition to current therapies such as corticosteroids. Here, we employed Illumina RNA sequencing and cytokine antibody proteome arrays to investigate how hypoxia modulates lung epithelial cell responses to HCoV-OC43 infection. The A549 lung epithelial cell line was used, given that alveolar type II epithelial cells are primary targets for SARS-CoV-2 replication in the lungs [29]. The A549 cell line has previously been used in studies involving SARS-CoV-2 and HCoV-OC43 [30,31]. HCoV-OC43 was used as a surrogate virus for SARS-CoV-2 as it belongs to the same genus and is well characterized, amenable to study and has been used to develop antiviral compounds targeting both HCoV-OC43 and SARS-CoV-2 in vitro, such as AT-527 [32], allowing us to investigate conserved innate immune responses that are important against a range of coronaviruses [33]. This approach could contribute towards the identification of immune mechanisms and potential therapeutic targets that could be of utility against both current and future emerging coronavirus infections.

## 2. Materials and Methods

### 2.1. Cell and Virus Culture

Human lung epithelial (A549) and fibroblast (MRC-5) cell lines were obtained from American Tissue Culture Collection (ATCC). These cell lines were maintained in Eagle’s Minimum Essential Medium (EMEM) with 10% fetal bovine serum (FBS), 100 units/mL of penicillin and 100 μg/mL of streptomycin (herein called growth media). Cells were cultured in 75 cm^2^ flasks with 10 mL of medium under normoxic conditions (20.9% O_2_, 5% CO_2_).

HCoV-OC43 VR-1558 was propagated as recommended (ATCC) in MRC-5 cells [34]. The virus was harvested by agitating glass beads in the flask to disrupt the cell monolayer, as described by [35]. The cell lysate was collected, aliquoted into cryovials, and stored at −80 °C. Viral titres were determined using a 50% tissue culture infectious dose (TCID_50_) assay, as described in [36].

### 2.2. Infection of A549 Cells with HCoV-OC43 Under Hypoxic and Normoxic Conditions

To investigate the effects of hypoxia and HCoV-OC43 infection on A549 cell inflammatory signaling, four conditions were selected. These were an uninfected normoxic control (N^Alone^), uninfected hypoxia alone (H^Alone^), normoxia + HCoV-OC43 (N^OC43^) and hypoxia + HCoV-OC43 (H^OC43^). A549 cells were seeded in 6-well plates at a density of 4 × 10^5^ cells per well in growth media and incubated overnight (24 h) prior to infection. Before infection, the medium was aspirated, and HCoV-OC43 was added at an MOI of 5, diluted in 500 µL of serum-free EMEM. The plates were incubated at 37 °C with 5% CO_2_ for 2 h to allow viral adsorption. Subsequently, 1.5 mL of growth media was added to each well, and the cells were incubated either under normoxic conditions (37 °C, 5% CO_2_, 20.9% O_2_) or in hypoxia (Baker Ruskinn InvivO_2_ 400 Physoxia Workstation; 37 °C, 5% CO_2_, 2% O_2_) for 24 h. Oxygen levels were independently verified using a separate Analox oxygen meter, in addition to the built-in oxygen sensor. mRNA was then isolated using the Quick-RNA Miniprep Kit (Zymo Research., Irvine, CA, USA cat: R1054) according to the manufacturer’s protocol.

### 2.3. RNA Sequencing of A549 Alveolar Epithelial Cells Infected with HCoV-OC43

The quality of isolated RNA was evaluated using a Qubit fluorometer (Thermo Fisher Scientific, Waltham, MA, USA) and an Agilent 2100 Bioanalyzer (Agilent Technologies, Inc., Santa Clara, CA, USA). RNA samples (*n* = 4) were then sent to Genewiz (Azenta Life Sciences, Germany) for library preparation and RNA sequencing, including poly-A enrichment. Sequencing was performed on an Illumina NovaSeq platform (version 2.0.0) with a 2 ×150 base pair read length and a depth of 20 million reads per sample. Raw reads were processed using Trimmomatic v0.36 for adapter removal and quality filtering. RNA sequencing data was mapped to the human reference genome (Homo sapiens GRCh38) so that the transcriptomic differences between conditions could be observed. Differentially expressed genes (DEGs) were identified using Differential Gene Expression Analysis Based on the Negative Binomial Distribution (DESeq2) [37], with significant DEGs defined by an absolute log2 fold change > 1 and an adjusted *p* value (padj) < 0.05 using the Wald’s test followed by a Benjamini–Hochberg correction for multiple comparisons. RNA sequencing data was also mapped to the HCoV-OC43 reference genome (ATCC VR-759-NC_006213.1) to assess the degree of infection between conditions by analyzing the relative abundance of structural genes [38,39]. Transcript per million (TPM) normalization was used to enable accurate quantitative comparisons of genes between conditions and among genes within the same condition [40].

### 2.4. Identification of Synergistic Genes

To identify genes with synergistic expression patterns, three DESeq2 comparisons were used: N^Alone^ vs. H^Alone^, N^Alone^ vs. N^OC43^, and N^Alone^ vs. H^OC43^. The comparisons were standardized to the same baseline condition, N^Alone^, to enable fold change comparisons. Genes were classified as synergistically regulated if the fold change in the H^OC43^ condition exceeded twice the sum of the fold changes observed in the individual conditions (synergistic downregulation was performed on the reciprocal of the fold changes), based on a modification of the approach of [41]. The DESeq2 analysis comparing N^OC43^ vs. H^OC43^ was used to determine whether changes in mRNA levels identified in the synergistic analysis were statistically significant between the two viral infection conditions. To assess whether synergistic gene expression was specific to A549 cells, MRC-5 lung fibroblasts were infected with HCoV-OC43 under hypoxic and normoxic conditions as described for A549 cells. Selected synergistic genes identified in A549 RNA-seq were analyzed by qRT-PCR to evaluate expression in a different cell type.

### 2.5. Gene Ontology Analysis

Significant DEGs were functionally annotated using ShinyGO v0.80 [42], using the list of all DEGs from the appropriate condition as a background. Gene ontology (GO) enrichment analysis was performed using the Kyoto Encyclopedia of Genes and Genomes (KEGG) and biological process (BP) databases. Significance was defined as a false discovery rate (FDR) of *p* < 0.05. Following enrichment, terms were filtered and sorted by significance (FDR), and then the top 10 terms were further sorted by fold enrichment to obtain the terms of interest.

### 2.6. Analysis of Cytokine and Immune-Related Protein Expression Using an Antibody Array

Expression of cytokines, chemokines and selected immune-related soluble proteins was quantified using the Proteome Profiler Human XL Cytokine Array (R&D Systems, Minneapolis, MN, USA, cat: ARY022B). 250 µL of cell culture supernatant from four independent biological repeats, each including all four experimental conditions, were pooled to create average (*n* = 4) samples of the four experimental conditions. Incubation and washing were carried out according to manufacturer’s instructions. Visualization of the cytokine arrays was performed using a LI-COR IRDye 800 CW streptavidin visualization antibody (LICOR, Lincoln, NE, USA, cat: 926-32230) and was performed according to the manufacturer’s instructions. Imaging of the arrays was performed on a LI-COR Odyssey M imager. Densitometry analysis was carried out on the cytokine arrays using LI-COR Image Studio Lite software version 5.2.5. To analyze cytokine expression between the four experimental conditions, each array was further normalized in relation to the average intensity of the reference spots on each array [43,44].

### 2.7. Quantification of IL-6 Concentration Using ELISA

IL-6 quantification was performed using the human IL-6 DuoSet ELISA Kit (R&D Systems, cat: DY206). Cell culture supernatants from four biological repeats were thawed and pooled to create an average sample of each condition. The ELISA was conducted according to manufacturer’s instructions. Absorbance was recorded at 450 nm using a BioTek Epoch 2 Microplate spectrophotometer, with 540 nm and 570 nm readings for correction. IL-6 concentrations were calculated using a 4-parameter logistic curve generated in R, aligning sample absorbances with the standard curve for accurate quantification.

### 2.8. Statistical Analyses

Statistical analyses were performed using R and GraphPad Prism v10. The specific statistical tests used are indicated in the figure legends. A *p*-value of <0.05 was considered statistically significant. RNA sequencing fold change data were rounded to the nearest whole number, while values exceeding 100 were rounded to the nearest ten. For cytokine array data, fold change values were rounded to one decimal place.

## 3. Results

### 3.1. Effect of HCoV-OC43 Infection on Lung Epithelial Cell Gene Expression in Hypoxia and Normoxia

To investigate the impact of hypoxia and HCoV-OC43 infection on gene expression in A549 cells, individually and in combination, RNA sequencing was used. Significant differentially expressed genes (DEGs), (defined as log2 fold change > 1 and an adjusted *p* value (padj) < 0.05) were identified using DESeq2 analysis [37] using three conditions, hypoxia alone (H^Alone^), normoxia + HCoV-OC43 (N^OC43^) and hypoxia + HCoV-OC43 (H^OC43^), all normalized to normoxia alone (N^Alone^). Cells incubated in the H^OC43^ condition exhibited a higher number of DEGS (4551), including both up- and downregulated mRNAs, compared to either stressor alone (H^Alone^, 1792 genes; N^OC43^, 2460 genes) (Figure 1A,B). Interestingly, in addition, the total number of significant DEGs was greater in the H^OC43^ condition (4551 genes) than the sum of the genes regulated by either stressor alone (H^Alone^ and N^OC43^) (4252 genes, Figure 1C), suggesting a synergistic response for a number of genes, which require both hypoxia and virus infection in order to be significantly differentially regulated. However, it must be borne in mind that the increase in DEGs seen in the H^OC43^ condition may in part reflect differences in mRNA stability in hypoxia [45] or due to virus infection, rather than solely changes in transcriptional activity.

Volcano plot (significance versus fold-change) mapping further supported these findings, highlighting the amplifying effect of dual stressors (hypoxia and HCoV-OC43 infection) on gene expression (Figure 1D–G). The hypoxic H^Alone^ condition caused 1792 genes to be significantly upregulated (blue) or downregulated (yellow), as shown in Figure 1D. The virus infection N^OC43^ condition (Figure 1E) induced more (2460) changes in gene expression, including a cluster of genes indicative of the activation of immune-related pathways, including upregulation of the mRNAs of proinflammatory genes, such as *IL6*, *CXCL8* and *IL1A*. The combined H^OC43^ condition (Figure 1F) had the most pronounced changes in gene expression, with dramatic increases in the number of both up- and downregulated genes compared to either condition alone. A direct comparison between H^OC43^ and N^OC43^ (Figure 1G) further confirmed that hypoxia significantly modulated cytokine genes during HCoV-OC43 infection identified in the N^OC43^ condition, with additional genes of interest being either upregulated or downregulated specifically under the hypoxia plus virus condition.

GO pathway enrichment analysis [42] provided insight into the biological processes modulated by each experimental condition by categorizing DEGs into different functional gene sets. As expected, in the H^Alone^ condition (Figure 2A), BP enriched terms were primarily related to cellular responses to hypoxia and epithelial cell regulation, while KEGG enriched pathways were associated with metabolic adaptations and hypoxic signaling, confirming that A549 cells were adapting to the hypoxic environment at the transcriptomic level. The N^OC43^ condition (Figure 2B) enriched GO terms and KEGG pathways associated with immune and inflammatory responses, including “response to lipopolysaccharide”, “inflammatory response”, and “cytokine-cytokine receptor interactions”, reflecting the activation of immune-related processes due to viral infection. In the H^OC43^ condition (Figure 2C), significant DEGs enriched pathways related to inflammatory responses, with these terms showing a much stronger and more significant enrichment compared to infection in normoxia (N^OC43^), with the number of genes in shared terms associated with inflammation, for example “inflammatory response”, increasing from 104 in N^OC43^ to 190 in infected cells in H^OC43^.

Together, these results confirm our finding of the effect of hypoxia and HCoV-OC43 infection in combination inducing a strong and diverse inflammatory and immune response, highlighting the amplifying role of hypoxia in modulating the cellular responses to viral infection.

### 3.2. Synergistic Effect of Hypoxia and HCoV-OC43 Infection on Synergistic Inflammatory Gene Expression

HCoV-OC43 infection in hypoxia appeared to exacerbate inflammatory responses compared to infection under normoxic conditions, as evidenced by GO and transcriptomic analyses (Figure 1 and Figure 2), suggesting a possible synergistic interaction between the two factors, as indicated previously in the literature [22]. Therefore, we decided to investigate this in more detail. We defined gene expression changes as synergistic when the fold change of a gene in the H^OC43^ condition was at least twice the sum of fold changes seen in the H^Alone^ and N^OC43^ conditions, normalized to N^Alone^. To identify synergistic downregulation, we applied the same principle to the reciprocal of the downregulated mRNA fold changes. In addition, we compared the results from these potential synergistically regulated genes with results from the N^OC43^ vs. H^OC43^ differential expression analysis to assess correlation and statistical significance between conditions. The analysis revealed 191 synergistically upregulated genes (Figure 3) and 43 synergistically downregulated genes (Figure 4). In total, 162 of these genes were protein-coding, 30 were long non-coding RNAs, 19 were long intergenic non-coding RNAs, 11 were pseudogenes, and 12 were unidentified transcripts. Complete data for all 191 synergistically regulated genes are provided in Appendix A. To determine whether synergistic gene expression was specific to the A549 cell line, MRC-5 lung fibroblast cells were infected with HCoV-OC43 under hypoxic or normoxic conditions using the same methodology described previously. Three synergistic genes, *IL1B, TNFAIP3* and *IRAK2* that exhibited synergistic upregulation in A549 cells and are key modulators of inflammatory signaling were then analyzed for expression in MRC-5 cells by qRT-PCR, confirming that all three genes were significantly upregulated in the H^OC43^ condition (Appendix A).

Key genes synergistically upregulated by HCoV-OC43 infection in hypoxia compared to individual conditions included *CSF2* (also known as *GM*-*CSF*), *IL6*, *CCL20*, *IL1A*, and *CXCL8* (also known as *IL-8*) (Figure 3). Some of the most differentially expressed genes, such as *CSF2* or *IL6*, had very large fold changes into the thousands. This is likely due to the *de novo* nature of expression, where the levels of mRNA at homeostasis were low or zero, resulting in very high fold-upregulation being observed. Levels of IL-6 protein have previously been shown to increase around 1000-fold during inflammatory events [46].

While HCoV-OC43 infection in hypoxia also induced synergistic downregulation, the overall fold change in these responses was smaller compared to the upregulated genes, and the downregulated genes appeared more random with no clear focus on immune or inflammatory pathways (Figure 4).

### 3.3. Effect of Hypoxia on HCoV-OC43 Induced Expression of Clinically Relevant Cytokine Genes

Severe COVID-19 cases, particularly those involving CS, are characterized by heightened levels of proinflammatory cytokines, which are critical predictors of disease severity and mortality [1]. We analyzed our A549 cell RNA sequencing data to investigate the expression of 24 cytokines commonly elevated in severe COVID-19, with a particular focus on their expression in epithelial cells and the influence of hypoxia. This panel of cytokines was selected based on multiple studies that analyzed serum cytokine levels in mild, severe or fatal COVID-19 cases, demonstrating elevated levels in more severely affected patients [7,8,9,10,11]. Our findings revealed that 10 out of the 24 cytokines, namely *CSF3*, *IL1A*, *IL1B*, *IL6*, *CXCL8*, *CCL2*, *CSF1*, *CCL20*, *PDGFB*, and *TNF,* were expressed at the RNA level by A549 lung epithelial cells. Eight of these ten cytokines were significantly upregulated in the H^OC43^ condition, compared to N^OC43^ (Figure 5). All genes were identified as synergistically expressed, apart from *CSF1* and *PDGFB.* This suggests that hypoxia may play a critical role in exacerbating the inflammatory response associated with COVID-19, potentially intensifying CS observed in severe cases. *PDGFB* gene expression was also significantly elevated in the H^Alone^ condition compared to N^OC43^ [47].

### 3.4. Effect of Hypoxia and HCoV-OC43 Infection on Cytokine Protein Secretion by A549 Lung Epithelial Cells

To investigate the impact of hypoxia and HCoV-OC43 infection on secreted cytokine protein expression in A549 cells, cytokine antibody arrays were used to analyze pooled cell culture supernatants (*n* = 4 independent biological repeats) from the A549 lung epithelial cell RNA sequencing experiments, in the four experimental conditions: N^Alone^, H^Alone^, N^OC43^, and H^OC43^. The arrays revealed distinct profiles of cytokine secretion depending on the treatment condition (Figure 6A). Biological significance was assumed if the fold change exceeded 1.5 [48].

Analysis of cytokine and immune-related protein expression showed that HCoV-OC43 infection in normoxia markedly upregulated proinflammatory proteins, including ICAM-1 (9.9-fold), IL-6 (13.0-fold), and IL-8 (8.0-fold) (Figure 6C). This pattern mirrored transcriptomic data from RNA sequencing (Figure 3). A similar profile of cytokine secretion was observed in the H^OC43^ condition, with IL-6 (15.3-fold) and IL-8 (8.1-fold) showing comparable increases to N^OC43^. Importantly, these results indicate that enhanced cytokine gene expression as determined by quantification of mRNA levels does not necessarily correlate directly to proportional increases in secreted proteins. IL-6 protein levels were quantified independently using a different technique (ELISA), confirming a strong induction in both N^OC43^ (1651.3 pg/mL) and H^OC43^ (1578.6 pg/mL) conditions (Figure 6F) with uninfected conditions showing no upregulation of IL-6.

Interestingly, there were elevated levels of IL-13, IL-4, and IL-5 across all conditions compared to N^Alone^, with fold changes exceeding the arbitrarily set threshold of biological significance (>1.50-fold). IL-13 levels increased 5.7-fold in H^Alone^, 6.3-fold in N^OC43^, and 6.3-fold in H^OC43^. IL-4 and IL-5 also displayed increases under all conditions (Figure 6B and Appendix A).

Only five of the cytokines, chemokines and immune-related proteins included in the array, namely Insulin-like Growth Factor Binding Protein-3 (IGFBP-3), Vascular Endothelial Growth Factor (VEGF), Chemokine (C-C motif) ligand 20 (CCL20), Cluster of Differentiation 30 (CD30) and myeloperoxidase (MPO), were specifically upregulated under H^OC43^ when compared to N^OC43^ above the set threshold of significance, with fold increases of 4.2, 2.1, 1.8, 2.0 and 1.8, respectively (Figure 6E). *CCL20* was also identified as a synergistically upregulated gene. These results suggest that while hypoxia and HCoV-OC43 infection can synergistically enhance the secretion of specific immune system proteins, this effect is selective and not uniform across all such proteins, even when synergy is observed at the transcriptomic level.

It is clear from the mRNA data presented in Figure 3, Figure 4 and Figure 5 and the protein data in Figure 6 that there is not a linear relationship between changes in mRNA level and protein level for the cytokine mRNA/protein combinations quantified. In order to investigate the relationship between mRNA and protein expression in hypoxia and during viral infection, the mRNAs coding for the top ten up- and downregulated cytokine and immune-related proteins identified from the antibody array in the H^OC43^ condition compared to the N^OC43^ were analyzed for the presence of putative internal ribosomal entry sites (IRESs) within their 5′UTRs using the Human IRES Atlas database [49,50]. Interestingly, analysis revealed that the mRNAs coding for seven out of the ten most upregulated proteins in H^OC43^ contain a putative IRES within the 5′UTR, compared to zero out of ten for the mRNAs of the downregulated cytokine and immune-regulated proteins identified in the array (Appendix A).

## 4. Discussion

Cytokine Storm (CS) has been implicated in increased mortality not only in COVID-19 [4] but also in many other severe inflammatory conditions such as bacterial sepsis [6]. Understanding the molecular mechanisms underlying CS is critical for developing targeted therapeutic strategies.

**Dif****ferentially expressed genes.** Analysis of our RNA sequencing data revealed that both hypoxia (H^Alone^) and HCoV-OC43 infection in normoxia (N^OC43^) individually caused changes in expression in a large number of genes, and additionally that hypoxia is able to strongly amplify signaling pathways activated by HCoV-OC43 infection, enhancing these responses in distinct ways (Figure 1 and Figure 2).

**Synergistic upregulation.** Importantly, the total number of differentially expressed genes (DEGs) was greater in the H^OC43^ condition (4551 genes) than the sum of the genes regulated by either stressor alone (H^Alone^ and N^OC43^) (4252 genes, Figure 1C), suggesting synergistic regulation (in which the fold change caused by hypoxia and virus together is more than double the sum of the fold changes caused by either factor alone) of a number of genes, involving hypoxia and virus infection combined.

Different modes of regulation between the two stimuli were observed. Some genes, such as *CSF2,* were not differentially regulated in H^Alone^, but the combination of hypoxia and viral infection resulted in a significantly higher level of CSF2 mRNA compared to N^OC43^ (160-fold in N^OC43^ compared to 7920-fold in H^OC43^, padj = 1.60 × 10^−5^, Figure 3). This indicates that hypoxia is able to strongly amplify signaling pathways activated by HCoV-OC43 infection. In contrast, some genes, such as *CCL20,* were differentially regulated in both separate conditions (16-fold in H^Alone^ and 1300-fold in N^OC43^) but exhibited much higher levels of mRNA in the combined H^OC43^ condition (2240-fold, padj = 2.05 × 10^−4^, Figure 3). This highlights a distinct synergistic effect where hypoxia and HCoV-OC43 infection-mediated signaling may converge to enhance the levels of gene expression beyond the levels seen in either condition alone. Importantly, there was also a group of purely synergistically regulated genes, including *TNF* and *TREM1* that showed no differential gene expression in H^Alone^ or N^OC43^ (Figure 3), indicating that these genes are only upregulated when both stimuli are present. Based on our data we propose three different modes by which hypoxia was observed to enhance inflammatory signaling:(1)amplification of pathways already activated by viral infection, where there is *no* upregulation by hypoxia alone, i.e., enhancement of virus-induced upregulation, as exemplified by *CSF2*,(2)amplification of pathways already activated by viral infection, where there *is* also upregulation by hypoxia alone, i.e., a combined effect on a gene also upregulated independently by both factors alone, as exemplified by *CCL20*,(3)increasing expression of genes which are not responsive to either stimulus alone, i.e., *bo**th* factors together are strictly required for significant upregulation, as exemplified by *TNF*.

**Synergistic** **downregulation**. The majority (191 of a total of 235 genes) of the synergistically regulated genes identified in the RNA sequencing data were upregulated. However, 44 genes were identified as being synergistically downregulated (Figure 4). Compared to the synergistically upregulated genes, the synergistically downregulated genes had lower fold changes and GO analysis revealed no specific biological emphasis compared to the inflammatory focus of the upregulated genes. Thus, while hypoxia and HCoV-OC43 infection together synergistically modulate proinflammatory genes, the downregulatory effects are associated with a wider range of biological processes. Interestingly this suggests purposeful positive upregulation of specific proinflammatory pathways as opposed to a more general non-specific downregulation of transcription across a wide range of genes, potentially even all genes, being responsible for relative downregulation of certain non-essential genes which may lack counter-balancing upregulation-promoting features such as binding sites for hypoxia-inducible transcription factors. This downregulation effect is likely geared towards energy conservation in hypoxia, which limits potential for ATP generation [49].

**Effects on specific cytokines and other inflammatory response proteins.** When compared with studies analyzing serum cytokine levels in patients with severe or fatal COVID-19 [7,8,9,10,11], our RNA sequencing data revealed synergistic upregulation of many of the same cytokines, with 10 out of the 24 proteins identified in clinical data also being synergistically upregulated (Figure 5). This suggests that coronavirus-infected epithelial cells, particularly under hypoxic conditions as a result of ARDS, may serve as a crucial contributor of excessive cytokine release contributing to a COVID-19 cytokine storm.

**CCL20.** When analyzing the specific cytokines, chemokines and immune-related proteins that were upregulated by HCoV-OC43 infection in hypoxia, our finding that CCL20 was upregulated in H^OC43^ compared N^OC43^ (Figure 6E) is particularly noteworthy and clinically relevant. CCL20 is a proinflammatory chemokine that primarily attracts T cells and dendritic cells to sites of infection and inflammation [51,52]. CCL20 interacts with its receptor CCR6 [53], which is particularly highly expressed on proinflammatory Th17 T cells and anti-inflammatory regulatory (Treg) T cells, causing migration and accumulation of these cells in lung tissue [54]. Analyses of bronchial lavage fluid from patients with severe COVID-19 have revealed a higher than normal proportion of Th17 cells relative to Treg cells, a pattern also observed in other lung injuries marked by hyperinflammation, such as ARDS and acute lung injury [55]. This excessive recruitment of Th17 cells, which secrete the important highly proinflammatory cytokine IL-17, contributes to the amplification of local inflammatory responses in the lungs, which plays a critical role in the development and progression of ARDS [56]. The persistent inflammatory signaling mediated by CCL20 contributes to the severity of pulmonary pathology [57]. This process can be further exacerbated by the fact that Th17 cells can themselves release CCL20, leading to a positive feedback loop.

From a mechanistic view, the *CCL20* gene has been shown to contain a HIF-1 responsive hypoxia response element (HRE) within the promoter, and *CCL20* gene expression has been shown to increase in chronic hypoxia [58] and *CCL20* was found to be synergistically upregulated by the combination of hypoxia and HCoV-OC43 infection in our RNA sequencing data (Figure 3). Hue et al. [59] found that serum levels of CCL20 were higher in patients with COVID-19 ARDS compared to patients with non-COVID-19 ARDS, and that the concentration increased over time, which was associated with prolonged ICU stays. This rise in serum CCL20 levels over time may be due in part to hypoxic signaling having a greater effect due to increased lung damage and ARDS linked to COVID-19 progression. This hypothesis is also supported by the fact that the same study reported the same trend in serum VEGF, a key hypoxia-related marker protein (as well as IL-8 which did not follow the same trend in our data) [16,59]. Serum levels of CCL20 were also found to be significantly higher in patients who died of COVID-19 compared to those that survived, indicating that CCL20 levels are an important negative predictor of survival in COVID-19 [11].

Therefore, targeting the CCL20/CCR6 axis could represent a promising future potential therapeutic opportunity for patients with severe COVID-19 who often exhibit elevated levels of Th17 cells [51,55]. At the time of writing, there are no currently approved drugs that target CCL20 or its target receptor CCR6. As CCL20/CCR6 mediates signaling through G-protein-coupled receptors [60], which are commonly and relatively easily targeted, our data suggest that this represents a promising potential future target for therapeutic intervention particularly in patients with severe, hypoxic COVID-19 disease. GSK3050002, a humanized IgG1κ antibody that binds to CCL20, has been investigated for use in inflammatory diseases [61] and this or a similar agent could potentially be effective in treating severe coronavirus infections.

**VEGF.** Using cytokine and immune-related protein arrays, A549 lung epithelial cells were observed to upregulate VEGF in H^Alone^ (1.8-fold increase compared to N^Alone^), with this expression being slightly higher in H^OC43^ (2.2-fold, Appendix A). When compared to N^OC43^, secreted VEGF protein expression was higher in H^OC43^, above the arbitrarily set threshold of biological significance (>1.5-fold, Figure 6E). Josuttis et al. [62] performed a retrospective screen of 139 COVID-19 patients, of which 71 had been admitted to ICUs. Patients requiring intensive care exhibited 1.9-fold higher protein levels of VEGF. Development of ARDS and organ failure could also be predicted by VEGF levels, with 277 pg/L on admission was correlated with increased risk of developing ARDS. VEGF is induced by hypoxia via the HIF-1 pathway [63] and contributes to inflammation in a number of ways. VEGF induces vascular permeability and vasodilation as well as increasing the expression of adhesion molecules such as ICAM-1 and VCAM-1 [64,65], with ICAM-1 being identified as a synergistic gene in our RNA sequencing data (Figure 3). Additionally, VEGF can act as a chemokine, recruiting monocytes to sites of infection [66]. Thus, under hypoxia, VEGF secretion by lung epithelial cells may drive increased inflammatory infiltration, increasing damage to the surrounding microenvironment and may contribute to ARDS and CS. Some clinical trials have already taken place using bevacizumab to target VEGF in the treatment of severe hypoxemic COVID-19, but the results have not yet been published [67].

**CD30 and MPO.** Antibody array data revealed that CD30 and myeloperoxidase (MPO) were both upregulated at the protein level in H^OC43^ compared N^OC43^, above the arbitrarily set threshold of biological significance (Figure 6E). Interestingly, RNA sequencing did not show differential expression of either mRNA, highlighting the importance of assessing protein levels as well as RNA levels to obtain an accurate picture of biological responses. CD30 is known to play a significant role in lung inflammation [68] and has been reported to be overexpressed in chronic lung conditions such as COPD [69]. However, the expression of CD30 and MPO in epithelial cells remains debated, as expression of these proteins is typically associated with immune cells [69,70]. Additionally, this observation may be driven by cell line selection, as lung cancer cell lines such as A549 cells have been shown to express CD30 [71], with no evidence of expression for MPO. Nevertheless, we have identified CD30 and MPO as interesting prospects for investigation as potential players in epithelial cell-mediated inflammatory responses.

**Other cytokines.** A surprising observation was the consistently elevated levels of IL-13, IL-5 and IL-4 proteins in H^Alone^, N^OC43^ and H^OC43^ compared to N^Alone^. Elevated IL-13 has been previously associated with severe COVID-19 outcomes, including the need for mechanical ventilation [72]. Despite the increased levels of IL-13, IL-4, and IL-5 under both hypoxia and HCoV-OC43 infection, there was no observed synergistic effect between these stimuli for these cytokines. However, hypoxia-induced cytokine expression in chronic conditions, such as COPD, may exacerbate inflammatory responses during coronavirus infections [73]. This suggests that patients with chronic lung hypoxia may be at increased risk of inflammatory complications due to elevated levels of IL-13, IL-5, and IL-4 released by the lung epithelium, possibly in combination with increased levels of other cytokines.

**Protective effects.** Although hypoxia-induced increases in proinflammatory cytokines may contribute to worse outcomes in patients with severe COVID-19, the data also suggest that hypoxia may exert context-dependent protective effects. Cytokine protein array analysis found that IGFBP-3 was upregulated 4.2-fold higher in H^OC43^ compared to N^OC43^ (Figure 6E). Lee et al. [74] demonstrated using a lung epithelial cell line and a mouse model of asthma that IGFBP-3 was able to inhibit inflammation by facilitating the activation of caspase 3/7 and caspase 8 which degraded IκBα and p65, reducing TNF-induced inflammation. Therefore, hypoxic signaling may play a protective role in controlling excess inflammation through the upregulation of IGFBP-3 in the lung. Furthermore, expression of certain synergistic genes, such PDGF-BB (Figure 5) has been shown to negatively correlate with the lung injury Murray score [7], suggesting a potential protective role of hypoxia in mitigating lung injury in COVID-19 patients. Therefore, these findings indicate that hypoxia may contribute to modulating immune responses during coronavirus infections by limiting tissue-damaging inflammation, highlighting new research opportunities.

**Disparity between gene and protein expression levels.** An interesting observation from our data was that relatively modest changes were seen in the expression of certain cytokine and immune-related inflammatory proteins compared to the respective increases in their mRNA levels in RNA sequencing data. This was especially notable in inflammatory genes identified as being synergistically regulated. *CSF2* was identified as a synergistic gene, being upregulated 160-fold in N^OC43^ and 7920-fold in H^OC43^ (Figure 3). However, at the protein level, CSF2 was only upregulated 3.5-fold in N^OC43^ and 3.7-fold in H^OC43^ (Appendix A). This drastic increase in the levels of *CSF2* mRNA resulted in little if any additional protein expression. This disparity between gene and protein expression has been reported before in hypoxia, e.g., for the matrix proteoglycan versican, where there was a >600-fold increase in mRNA but a only 3-fold increase in protein expression in primary human macrophages [75]. In hypoxia, some proteins have been reported to be preferentially expressed through different mechanisms, such as Internal Ribosome Entry Sites (IRES), upstream open reading frames (uORFs) or selective mRNA partitioning [49]. IRES are crucial in hypoxia because they allow cells to bypass the usual 5′ cap-dependent translation pathway, which is often inhibited during low oxygen conditions [76]. Analysis of our proteomic data using the Human IRES Atlas found that proteins that were upregulated in H^OC43^ (compared to N^OC43^) were more likely to contain an IRES compared to those that were downregulated by the same comparison (Appendix A) [50]. CCL20 which was identified as a synergistically regulated mRNA (Figure 3) and which underwent a 1.8-fold increase in protein expression in hypoxia (Figure 6E) was also identified as containing an IRES, whereas as other synergistically regulated cytokine mRNAs such as *IL6* or *CSF2* without a predicted IRES underwent little to no upregulation at the protein level (1.2-fold and 1.0-fold). Thus, the increased expression of certain proinflammatory cytokines during coronavirus infections in hypoxia could be, at least in part, dependent on the presence of an IRES within the 5′UTR of the mRNA, and experimental investigation of this interesting possibility which our data highlight represents an exciting future avenue of investigation. Both CD30 and MPO proteins were upregulated above the threshold in H^OC43^ (compared to N^OC43^), and were predicted to contain a putative IRES, so this may have facilitated the increased expression of these proteins in hypoxia during HCoV-OC43 infection, even when no differential expression was seen in RNA sequencing data. However, it is important to note that the presence of outliers within the data set, such as IGFBP-3, indicates that while potentially important, this could be one of several contributing factors and is unlikely to be the single deciding factor for all genes. In summary, both in the present study and across the wider literature, the exact reasons for disparity between mRNA levels and protein levels in hypoxia for particular genes remain to be confirmed. It may simply be that increased levels of specific mRNAs are required to ensure continued production of or modest increases in protein levels, given the energetic and consequent translational inhibition constraints imposed by hypoxia.

The use of HCoV-OC43 in this study is grounded in practical and mechanistic considerations. HCoV-OC43 shares several key features with SARS-CoV-2, including aspects of genomic organization, replication, and host immune modulation, making it a suitable surrogate for studying conserved coronavirus-host interactions. HCoV-OC43 typically causes mild upper respiratory illness and does not typically cause the severe pulmonary pathology associated with early strains of SARS-CoV-2 [77]. However, more recently identified SARS-CoV-2 variants, such as Omicron and its sublineages, have demonstrated a shift in tropism toward the upper respiratory tract [78], aligning more closely with HCoV-OC43 infection. This apparent convergence strengthens the use of HCoV-OC43 as a comparative model, but care should be taken as to not extrapolate findings without further validation. While HCoV-OC43 is a virus with lower pathogenicity compared to SARS-CoV-2, molecular epidemiologic evidence indicates that this virus arose via a zoonotic transmission event, possible moving into the human population from cattle, around 1890 [79]. It has been speculated, although not proven, that this virus could have been responsible for the “Russian Flu” pandemic which occurred in 1889–1890 and caused at least 1 million excess deaths worldwide [80]. The current low pathogenicity of the now-endemic HCoV-OC43 virus may also in part be due to exposure earlier in life, as children can respond very differently to viruses compared to adults, as seen for COVID-19 [81].

In summary, we have shown that hypoxia (H^Alone^) and HCoV-OC43 infection in normoxia (N^OC43^) caused changes in expression in a large number of genes, and that there are synergistic increases in expression of certain genes in HCoV-OC43 infection in hypoxia. There are corresponding changes in protein expression levels (<13×) but these are much less than the increase in mRNA levels (<8000). Proteins that were upregulated in H^OC43^ (compared to N^OC43^) were more likely to contain an IRES within the 5′UTR of the mRNA compared to those that were downregulated by the same comparison. The main applications of this work relate to the observation that increased expression is largely related to proinflammatory pathways, and that some changes in expression are related to protective mechanisms, both of which may be relevant to the search for therapeutic interventions. To complement and expand the in vitro data presented here, extension of the work into animal models, primary cell cultures and organoid models could further enhance our understanding of how hypoxia modulates inflammatory responses to HCoV-OC43 infection. In addition to hypoxia, hyperoxia could be included in future studies since supplemental oxygen and even hyperbaric oxygen supplementation is often used in the treatment of respiratory virus and other airway conditions.

## 5. Conclusions

Our study highlights inflammatory changes in mRNA and protein levels that distinguish hypoxic and normoxic HCoV-OC43-infected lung epithelial cells and identifies possible regulatory mechanisms for further investigation. Our work also identifies two potential therapeutic targets, VEGF and CCL20, which are synergistically upregulated by hypoxia during HCoV-OC43 infection, the targeting of which, for example, by modulating hypoxia signaling pathways such as the HIF-1 pathway which regulates both these proteins, may be of utility in controlling hyperinflammation and inhibiting cytokine storm in patients with severe COVID-19.

## Figures and Tables

**Figure 1 biomolecules-15-01144-f001:**
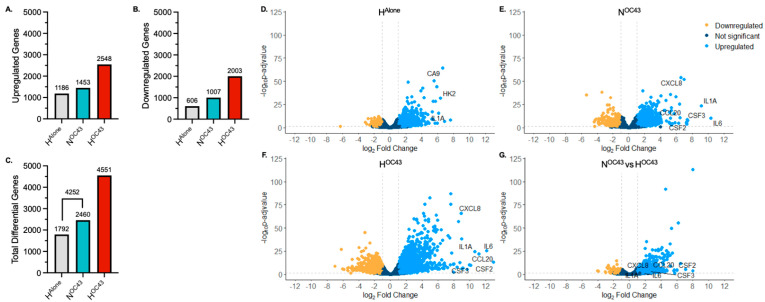
Transcriptomic Profiling of RNA Sequencing Data from A549 Lung Epithelial Cells. (**A**–**C**), Variation in significantly differentially expressed genes (DEGs) between RNA sequencing readouts from cells incubated in different experimental conditions (*n* = 4). The x-axis shows the condition group, and the y-axis represents the number of significantly (**A**) upregulated, (**B**) downregulated and (**C**) total DEGs compared to N^Alone^. Differential gene expression was analyzed using the DESeq2 R package. A gene was defined as significantly differentially expressed if *p* ≤ 0.05 (using the Wald test followed by a Benjamini–Hochberg correction for multiple testing) and if the log2 fold change was >1. (**D**–**G)**, Volcano plots showing the distribution of DEGs in A549 cells in (**D**) H^Alone^, (**E**) N^OC43^ and (**F**) H^OC43^, compared to N^Alone^. Volcano plot (**G**) shows DEGs in H^OC43^, compared to N^OC43^. Clinically relevant cytokine genes of interest are annotated in each volcano plot. The x-axis shows the log2 fold change in each gene and the y-axis shows the log10 (*p*-adjusted value) of each gene. Genes with a *p*-value less than 0.05 and a log2 fold change greater than 1 are indicated by blue dots. These represent upregulated genes. Genes with a *p*-value less than 0.05 and a log2 fold change less than −1 are indicated by yellow dots. These represent downregulated genes. Genes with a *p*-value greater than 0.05, or a log2 fold change less than one or greater than −1 are indicated by dark blue dots. These represent unchanged genes. All data is based on *n* = 4 independent biological repeats.

**Figure 2 biomolecules-15-01144-f002:**
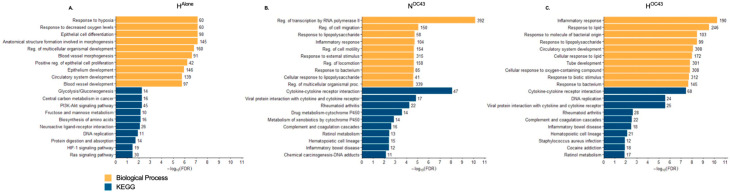
Gene Ontology of RNA Sequencing Data from A549 Lung Epithelial Cells. (**A**–**C**), GO clustering of significantly DEGs in A549 lung epithelial cell RNA sequencing data. GO analysis was performed using ShinyGO v0.80, focusing on the BP and KEGG GO term databases. The analysis included DEGs under the three conditions: (**A**) H^Alone^, (**B**) N^OC43^, and (**C**) H^OC43^, each compared to N^Alone^. The top ten GO terms identified through this filtering and ranking process are displayed per condition. The x-axis shows the −log10 of the FDR (False Discovery Rate) values, with the more significant pathways having longer bars, per GO term. GO terms were filtered based on their false discovery rate (FDR) and then ranked by fold enrichment. Numbers at the end of each column indicate the number of genes regulated. All data is based on *n* = 4 biological repeats.

**Figure 3 biomolecules-15-01144-f003:**
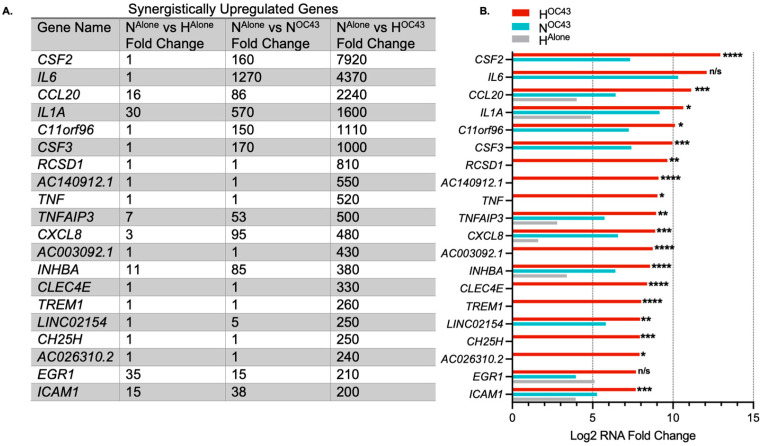
The most highly synergistically upregulated genes induced by HCoV-OC43 infection of A549 cells in hypoxia. (**A**) The top 20 genes showing the highest synergistic upregulation in response to HCoV-OC43 infection under hypoxic conditions (*n* = 4). For this study, synergistic gene upregulation was defined as upregulation more than two-fold higher in the presence of both hypoxia and HCoV-OC43 infection than the sum of fold changes by either condition alone. The fold changes shown are for H^Alone^, N^OC43^, and H^OC43^, normalized to N^Alone^. In total, 191 genes were identified as synergistically upregulated, including 133 protein-coding genes, 25 long non-coding RNAs (lncRNAs), 12 long intergenic non-coding RNAs (lincRNAs), 9 pseudogenes, and 12 undefined transcripts. (**B**) The bar chart displays the log2-transformed fold changes in mRNA levels for the top 20 synergistically upregulated genes in H^OC43^ (red), N^OC43^ (turquoise) and H^Alone^ (gray). Statistical significance between N^OC43^ and H^OC43^ was assessed using a Wald test followed by Benjamini–Hochberg correction for multiple testing based on the DESeq2 differential analysis. Significance levels are indicated as follows: n/s = not significant, * = *p* < 0.05, ** = *p* < 0.01, *** = *p* < 0.001, **** = *p* < 0.0001. Complete data for all 191 synergistically upregulated genes are provided in Appendix A.

**Figure 4 biomolecules-15-01144-f004:**
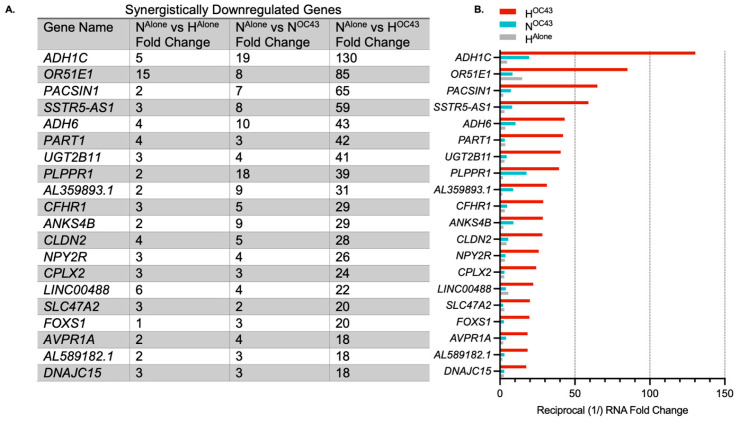
Top synergistically downregulated genes induced by HCoV-OC43 infection of A549 cells in hypoxia. (**A**) The top 20 genes showing the highest synergistic downregulation in response to HCoV-OC43 infection under hypoxic conditions (*n* = 4). For this study, synergistic gene downregulation was defined as a reduction in the number of transcripts by more than two-fold in the presence of both hypoxia and HCoV-OC43 infection than the sum of fold changes by either condition alone. The fold changes shown are for H^Alone^, N^OC43^, and H^OC43^, normalized to N^Alone^. In total, 43 genes were identified as synergistically downregulated, including 29 protein-coding genes, 5 lncRNAs, 7 lincRNAs, and 2 pseudogenes. (**B**) The bar chart displays the log2-transformed fold changes in mRNA levels for the top 20 synergistically downregulated genes in H^OC43^ (red), N^OC43^ (turquoise) and H^Alone^ (gray). Statistical significance between N^OC43^ and H^OC43^ was assessed using a Wald test followed by Benjamini–Hochberg correction for multiple testing based on the DESeq2 differential analysis, but no genes were statistically significant. Complete data for all 44 synergistically regulated genes are provided in Appendix A.

**Figure 5 biomolecules-15-01144-f005:**
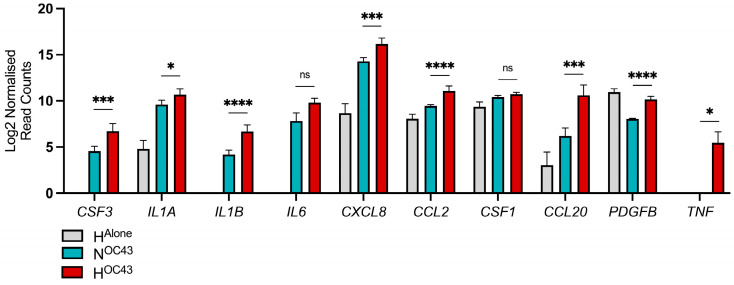
Transcriptomic analysis of clinically relevant cytokine genes expressed by A549 cells. RNA sequencing data (*n* = 4) from A549 lung epithelial cells infected with HCoV-OC43 under normoxic and hypoxic conditions were compared to a curated list of clinically relevant cytokines identified from studies comparing serum cytokine levels in severely and mildly affected COVID-19 patients. A panel of cytokines was selected based on multiple studies that analyzed serum cytokine levels in COVID-19 patients with mild, severe or death cases, demonstrating elevated levels in those with more severe disease [7,8,9,10,11]. These cytokines were compared to the RNA sequencing data to assess whether hypoxia influences their levels of mRNA in lung epithelial cells during HCoV-OC43 infection. The displayed data represents the log2-transformed reads for each gene in H^Alone^ (gray), N^OC43^ (turquoise), and H^OC43^ (red), normalized to N^Alone^. Statistical significance of differential expression between N^OC43^ and H^OC43^ was assessed using a Wald test followed by Benjamini–Hochberg correction for multiple testing, based on the DESeq2 differential analysis: ns = not significant, * = *p* < 0.05, *** = < 0.001, **** = < 0.0001.

**Figure 6 biomolecules-15-01144-f006:**
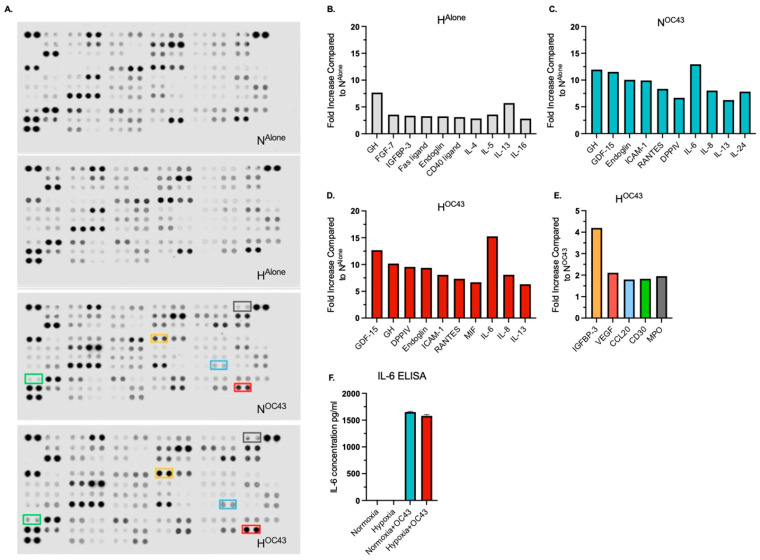
Secreted/released cytokine, chemokine and immune-related protein expression in A549 lung epithelial cells infected with HCoV-OC43 under normoxic and hypoxic conditions. Protein levels were assessed using cytokine, chemokine and immune-related antibody arrays, analyzing pooled cell culture supernatant samples from the RNA sequencing experiments (*n* = 4 independent biological repeats). (**A**) shows the antibody arrays used to evaluate protein expression across the four experimental conditions. (**B**–**D**) display the top 10 upregulated protein in each condition, normalized to N^Alone^. (**E**) highlights the expression of the five proteins—IGFBP-3, VEGF, CCL20, CD30 and MPO—that were upregulated by more than 1.5-fold in the H^OC43^ condition compared to the N^OC43^ condition. The color of the bars corresponds to the color-coded boxes on the antibody arrays. (**F**) IL-6 ELISA assay on pooled supernatant samples (*n* = 4).

## Data Availability

The original contributions presented in this study are included in the article/Appendix A. Further inquiries can be directed to the corresponding authors.

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
