# Peer review of "Hypoxia Exacerbates Inflammatory Signaling in Human Coronavirus OC43-Infected Lung Epithelial Cells"

_biomolecules, 2025, doi:10.3390/biom15081144_

Round 1

Reviewer 1 Report

Comments and Suggestions for Authors

Manuscript ID. biomolecules-3736977

Hypoxia Exacerbates Inflammatory Signaling in Coronavirus-2 Infected Lung Epithelial Cells. Jarod Zvartau-Hind, Hassan Sadozai, Hateem Z. Kayani, Animesh Acharjee, Rory Williams, Phillip Gould, Christopher A. Reynolds, and Bernard Burke.

I only have some constructive comments that can improve this article.

  1. In the introduction, it would be helpful to add a paragraph mentioning the functional repercussions of hypoxia in the context of conditions such as asthma, chronic obstructive sleep apnea, lung cancer, idiopathic pulmonary fibrosis, obstructive sleep apnea, influenza, and COVID-19. This is important because hypoxia plays an important role in the pathophysiology of chronic and acute respiratory diseases.
  2. How was it ensured that hypoxic conditions (2% O₂) were maintained consistently throughout the experiment? The methodology does not mention the use of an oxygen monitoring system, such as a specific analyzer or sensor, for example, the Teledyne 60T or OOM105.
  3. At any point was the inclusion of a hyperoxia experimental condition considered as an additional or comparative control? This is especially relevant since the cytokine storm (CS) has been implicated not only in COVID-19 but also in other severe inflammatory conditions.
  4. Wouldn't it be relevant to include a hyperoxia experimental condition? Considering that during the COVID-19 pandemic, non-pharmacological interventions such as hyperbaric oxygen therapy were used, which showed positive effects in modulating the inflammatory and antithrombotic response, in addition to reducing tissue hypoxia, the production of viricidal reactive oxygen intermediates, and the modulation of stem cells and cytokines.
  5. Could you clarify whether MRC-5 fibroblasts were used solely for the propagation of the HCoV-OC43 virus and not involved in subsequent transcriptomic and proteomic analyses? If they were used in any other phase of the study, we would appreciate a clearer specification of the methodology.
  • Although I see that they appear in the results as a control to determine whether the synergistic gene expression was specific to the A549 cell line.
  1. The results are truly compelling and interesting. This study primarily used A549 alveolar epithelial cells and was validated in MRC-5F pulmonary fibroblasts. The synergistic expression of key inflammatory genes suggests an amplification of the inflammatory response. This reinforces the biological significance of the findings and suggests a possible common response across different lung cell types. However, it is important to recognize that in vitro cell models have limitations, as they fail to replicate the complexity of the tissue microenvironment or the interaction with other cell types and systemic signals present in a living organism. It would be important, in the future, to complement these studies in animal models and expand our understanding of the inflammatory mechanisms of HCoV-OC43 and their modulation by hypoxia.
  2. Regarding the discussion, I find it very well structured and enjoyable to read. However, it is important to note that the study demonstrates an exacerbated inflammatory response under hypoxic conditions during HCoV-OC43 infection. Unfortunately, the discussion does not address the central role played by hypoxia-inducible transcription factors, particularly HIF-1α and HIF-2α. These regulators are activated under low oxygen concentration conditions, such as those observed in infected lungs, and a key opportunity to delve deeper into the molecular mechanisms involved is missed.
  3. Finally, in conclusion, I believe caution should be exercised when extrapolating the results of an in vitro cellular model to the pathophysiology of the disease.

Author Response

Reviewer 1 Comments –

  1. In the introduction, it would be helpful to add a paragraph mentioning the functional repercussions of hypoxia in the context of conditions such as asthma, chronic obstructive sleep apnea, lung cancer, idiopathic pulmonary fibrosis, obstructive sleep apnea, influenza, and COVID-19. This is important because hypoxia plays an important role in the pathophysiology of chronic and acute respiratory diseases.

We thank the reviewer for this helpful suggestion and have added a paragraph covering this to the Introduction starting at line 76:

“Hypoxia has been identified as a risk factor in various chronic and acute airway diseases. For example, chronic intermittent hypoxia, such as that observed in obstructive sleep apnea, has been shown to exacerbate airway hyperresponsiveness, elevate the expression of proinflammatory cytokines, and promote immune cell infiltration in in vivo models of asthma [26,27]. Moreover, infection of epithelial cells with an Influenza H1N1 virus under hypoxic conditions leads to increased expression of the proinflammatory cytokines TNF and IL-6, and a decrease in the anti-inflammatory cytokine IL-10 [23].”

  1. How was it ensured that hypoxic conditions (2% O₂) were maintained consistently throughout the experiment? The methodology does not mention the use of an oxygen monitoring system, such as a specific analyzer or sensor, for example, the Teledyne 60T or OOM105.

The Baker Ruskinn InvivO₂ 400 Physoxia Workstation used is a state-of-the-art instrument and has an inbuilt highly accurate oxygen sensor which continuously senses and controls the level of oxygen within the chamber. Nevertheless, the oxygen level was verified using a second ANALOX oxygen meter, which is independent of the InvivO₂ 400 instrument. This useful point has been added to the Methods section starting from line 124.

“Oxygen levels were independently verified using a separate Analox oxygen meter, in addition to the built-in oxygen sensor.”

  1. At any point was the inclusion of a hyperoxia experimental condition considered as an additional or comparative control? This is especially relevant since the cytokine storm (CS) has been implicated not only in COVID-19 but also in other severe inflammatory conditions.

We thank the reviewer for this interesting and useful suggestion. Those particular experiments were not performed in the present study, but this suggestion has now been mentioned in the Discussion section as a suggestion for future work in this area in lines 661-663.

“In addition to hypoxia, hyperoxia could be included in future studies since supplemental oxygen and even hyperbaric oxygen supplementation is often used in the treatment of respiratory virus and other airway conditions.”

  1. Wouldn't it be relevant to include a hyperoxia experimental condition? Considering that during the COVID-19 pandemic, non-pharmacological interventions such as hyperbaric oxygen therapy were used, which showed positive effects in modulating the inflammatory and antithrombotic response, in addition to reducing tissue hypoxia, the production of viricidal reactive oxygen intermediates, and the modulation of stem cells and cytokines.

See above response to point 3. In addition, it can be difficult to know exactly what super-physiological oxygen concentration should be used in such in vitro experiments to mimic the hyperbaric oxygen therapy situation. However, this is an interesting constructive suggestion for future studies and has been included in the Discussion as described in point 3 above, lines 661-663.

“In addition to hypoxia, hyperoxia could be included in future studies since supplemental oxygen and even hyperbaric oxygen supplementation is often used in the treatment of respiratory virus and other airway conditions.”

  1. Could you clarify whether MRC-5 fibroblasts were used solely for the propagation of the HCoV-OC43 virus and not involved in subsequent transcriptomic and proteomic analyses? If they were used in any other phase of the study, we would appreciate a clearer specification of the methodology.
  • Although I see that they appear in the results as a control to determine whether the synergistic gene expression was specific to the A549 cell line.

To address and clarify this valid point, the following has been added to section 2.4 in the Methods section in lines 154-158.

To assess whether synergistic gene expression was specific to A549 cells, MRC-5 lung fibroblasts were infected with HCoV-OC43 under hypoxic and normoxic conditions as described for A549 cells. Selected synergistic genes identified by A549 RNA-seq were analyzed by qRT-PCR to evaluate expression in a different cell type.

  1. The results are truly compelling and interesting. This study primarily used A549 alveolar epithelial cells and was validated in MRC-5F pulmonary fibroblasts. The synergistic expression of key inflammatory genes suggests an amplification of the inflammatory response. This reinforces the biological significance of the findings and suggests a possible common response across different lung cell types. However, it is important to recognize that in vitro cell models have limitations, as they fail to replicate the complexity of the tissue microenvironment or the interaction with other cell types and systemic signals present in a living organism. It would be important, in the future, to complement these studies in animal models and expand our understanding of the inflammatory mechanisms of HCoV-OC43 and their modulation by hypoxia.

Another useful point. A sentence covering the need for in vivo models has been added to the Discussion, lines 658-661.

To complement and expand the in vitro data presented here, extension of the work into animal models, primary cell cultures and organoid models could further enhance our understanding of how hypoxia modulates inflammatory responses to HCoV-OC43 infection.

  1. Regarding the discussion, I find it very well structured and enjoyable to read. However, it is important to note that the study demonstrates an exacerbated inflammatory response under hypoxic conditions during HCoV-OC43 infection. Unfortunately, the discussion does not address the central role played by hypoxia-inducible transcription factors, particularly HIF-1α and HIF-2α. These regulators are activated under low oxygen concentration conditions, such as those observed in infected lungs, and a key opportunity to delve deeper into the molecular mechanisms involved is missed.

This is a very interesting point and indeed a detailed dissection of the role of HIFs in the responses identified could be an area for future study. This is complicated by the fact that HIFS are regulated both by oxygen levels (at the level of protein stability) and also (via a different mechanism, transcriptional regulation) by inflammatory stimuli such as pathogen associated molecular patterns (PAMPS) including lipopolysaccharide. Nevertheless the roles of HIFs in regulating some of the key hypoxia-regulated genes identified has been discussed in lines 509, 543, and 668.

  1. Finally, in conclusion, I believe caution should be exercised when extrapolating the results of an in vitro cellular model to the pathophysiology of the disease.

A valid point which has now been specifically addressed in the additional text added to the Discussion (see review point 6 above) in lines 658-661 covering the need to use in vivo models to further validate the in vitro findings of this study.

To complement and expand the in vitro data presented here, extension of the work into animal models, primary cell cultures and organoid models could further enhance our understanding of how hypoxia modulates inflammatory responses to HCoV-OC43 infection.

In addition, the Conclusions wording has been toned down. This section (lines 665-673) now reads:

“Conclusions

Our study highlights inflammatory changes in mRNA and protein levels that distinguish hypoxic and normoxic HCoV-OC43-infected lung epithelial cells, and identifies possible regulatory mechanisms for further investigation. Our work also identifies two potential therapeutic targets, VEGF and CCL20, which are synergistically-upregulated by hypoxia during HCoV-OC43 infection, targeting of which, for example by modulating hypoxia signaling pathways such as the HIF-1 pathway which regulates both these proteins, may be of utility in controlling hyperinflammation and inhibiting cytokine storm in patients with severe COVID-19.”

Reviewer 2 Report

Comments and Suggestions for Authors

The current manuscript under review, titled “Hypoxia Exacerbates Inflammatory Signaling in Coronavirus-Infected Lung Epithelial Cells” explores the synergistic effect of hypoxia and coronavirus OC43 (HCoV-OC43) infection on the inflammatory response in lung epithelial cells (A549). The authors have reported by transcriptomic (RNA-seq) and proteomic (antibody arrays, ELISA) profiling that hypoxia exacerbates virus-induced inflammatory state in these cells. They show a significant increase in the proinflammatory cytokine expression at both mRNA and protein levels. Pro-inflammatory cytokines like IL-6, VEGF, and CCL20 were reported to be higher in the hypoxia + viral-infection group, suggesting that hypoxia exacerbates cytokine storm (CS) risk in SARS-CoV-2 infection, and CCL20 and VEGF as potential therapeutic targets.

The authors are applauded for their efforts and for conducting a detailed study to validate their hypothesis. This study is clinically relevant as it explores one of the key aspects of SARS-CoV-2 infection, i.e., hypoxia, which is a very common symptom in severe cases. The manuscript has used rigorous statistical techniques across all experiments. The novelty of the study lies in showing a synergistic interaction between hypoxia and SARS virus infection. However, I would like to highlight some of my concerns that I believe will strengthen the manuscript.

  1. It’s a nice idea to make the manuscript titles, but one needs to be cautious not to create a bias or miss to reflect the actuality of the study. I suggest changing the title from coronavirus to specifically mention coronavirus OC43 (HCoV-OC43).
  2. OC43(HCoV-OC43), despite being a part of the coronavirus family, has much less severe consequences of infection, e.g., it is less likely to progress to cause ARDS. It infects the upper respiratory tract, unlike SARS-CoV-2. The reason I am highlighting this point is because the authors have heavily banked on the SARS-CoV-2, ARDS and cytokine storm, but such phenomena are not very well established after OC43 infections. How do authors justify the extrapolation of data? I would suggest discussing it thoroughly in the discussion while clearly stating the limitations of the model used in the current study. The use of OC43 weakens translational claims for COVID-19.
  3. A549 indeed are epithelial cells of lung origin and undoubtably have been extensively used in lung biology research. However, we cannot ignore the fact that these cells are cancerous. Cancerous cells, unlike normal cells, have either an immunosuppressive or pro-inflammatory phenotypes. How can authors address the extrapolation of the observed data to link with the response observed in post SARS-CoV-2 infection where we are linking a cancerous epithelial cell with normal human lung cells? Moreover, both A549and MRC-5 are not true representations of primary pulmonary cells. Why did authors prefer A549 cells over using primary pulmonary epithelial cells?
  4. The manuscript observes a notable disconnect between mRNA and protein expression (e.g., IL-6, CSF2), attributing this to IRES elements. How would the authors choose or address a validation of this discrepancy.

Minor concerns

  1. The suggestion of targeting VEGF or CCL20 for therapy, though biologically interesting, lacks in vivo support or pharmacologic data. Tone this down or frame as speculative.
  2. The term “synergistic regulation” is used throughout. Authors should define the exact statistical threshold more clearly upfront and consistently refer to it.
  3. Emphasize limitations of using HCoV-OC43 in translational relevance.
  4. Consider additional validation in primary cells or organoid models.
  5. Provide experimental support for IRES hypothesis or tone down conclusions.
  6. Streamline figures and reorganize supplementary data for clarity.

Author Response

Reviewer 2 Comments –

  1. It’s a nice idea to make the manuscript titles, but one needs to be cautious not to create a bias or miss to reflect the actuality of the study. I suggest changing the title from coronavirus to specifically mention coronavirus OC43 (HCoV-OC43).

We thank the reviewer for this very useful point. This change to the title has been made.

  1. OC43(HCoV-OC43), despite being a part of the coronavirus family, has much less severe consequences of infection, e.g., it is less likely to progress to cause ARDS. It infects the upper respiratory tract, unlike SARS-CoV-2. The reason I am highlighting this point is because the authors have heavily banked on the SARS-CoV-2, ARDS and cytokine storm, but such phenomena are not very well established after OC43 infections. How do authors justify the extrapolation of data? I would suggest discussing it thoroughly in the discussion while clearly stating the limitations of the model used in the current study. The use of OC43 weakens translational claims for COVID-19.

This is an important point, and care should be taken not to over-extrapolate the data to other coronavirus infections other than the HCoV-OC43 virus that was used. An extensive paragraph covering the use of HCoV-OC43 has been added to the discussion addressing these points, starting from line 630-647.

“The use of HCoV-OC43 in this study is grounded in practical and mechanistic considerations. HCoV-OC43 shares several key features with SARS-CoV-2, including aspects of genomic organization, replication, and host immune modulation, making it a suitable surrogate for studying conserved coronavirus-host interactions. HCoV-OC43 typically causes mild upper respiratory illness and does not typically cause the severe pulmonary pathology associated with early strains of SARS-CoV-2. However, more recently identified SARS-CoV-2 variants, such as Omicron and its sublineages, have demonstrated a shift in tropism toward the upper respiratory tract, aligning more closely with HCoV-OC43 infection. This apparent convergence strengthens the use of HCoV-OC43 as a comparative model, but care should be taken as to not extrapolate findings without further validation.  While HCoV-OC43 is a virus with lower pathogenicity compared to SARS-CoV-2, molecular epidemiologic evidence indicates that this virus arose via a zoonotic transmission event, possible moving into the human population from cattle, around 1890. It has been speculated, although not proven, that this virus could have been responsible for the “Russian Flu” pandemic which occurred in 1889-1890 and caused at least 1 million excess deaths worldwide. The current low pathogenicity of the now-endemic HCoV-OC43 virus may also in part be due to exposure earlier in life, as children can respond very differently to viruses compared to adults, as seen for COVID-19.”

  1. A549 indeed are epithelial cells of lung origin and undoubtably have been extensively used in lung biology research. However, we cannot ignore the fact that these cells are cancerous. Cancerous cells, unlike normal cells, have either an immunosuppressive or pro-inflammatory phenotypes. How can authors address the extrapolation of the observed data to link with the response observed in post SARS-CoV-2 infection where we are linking a cancerous epithelial cell with normal human lung cells? Moreover, both A549 and MRC-5 are not true representations of primary pulmonary cells. Why did authors prefer A549 cells over using primary pulmonary epithelial cells?

We thank the reviewer for these points, which are valid and important. The use of cell lines is indeed far from optimal but was the only feasible option for us at the time that this part of the study was carried out, during the COVID-19 pandemic 2020-2022. It was unfortunately not possible for us to obtain primary human lung epithelial cells during this period, and it is rarely possible to obtain these at the best of times. Use of non-human primary cells could have been attempted but this also has drawbacks since the virus being used infects humans, and of course we wished to study the responses of human cells to coronavirus infection in hypoxia. An additional sentence has been added to the discussion (see review point 6 above) from lines 658-661 to address the issue of the importance of validating the work with the use of a range of different model systems.

To complement and expand the in vitro data presented here, extension of the work into animal models, primary cell cultures and organoid models could further enhance our understanding of how hypoxia modulates inflammatory responses to HCoV-OC43 infection.

  1. The manuscript observes a notable disconnect between mRNA and protein expression (e.g., IL-6, CSF2), attributing this to IRES elements. How would the authors choose or address a validation of this discrepancy.

The reviewer has rightly identified an interesting observation in our study, and this is certainly an area for potential further studies. This discrepancy between mRNA levels and protein levels in a range of cells is a previously reported and widely acknowledged one, and a reference is given within the paper to support this (Sotoodehnejadnematalahi et al, line 602 [75]). Further investigation of this is of interest but since it lies outside the main areas of expertise of our team at present, we hope that this will be followed up at a later date after publication of our initial findings.

Minor concerns

  1. The suggestion of targeting VEGF or CCL20 for therapy, though biologically interesting, lacks in vivo support or pharmacologic data. Tone this down or frame as speculative.

We thank the reviewer for this valid point, and the abstract (line 27) and the discussion section (starting from line 526) covering this has been toned down and the wording in the relevant paragraphs has been changed to be more speculative. It is noteworthy that both these molecules are well known to have important biological roles and have been proposed as therapeutic targets in a variety of pathogenic conditions, including in the lung.

In addition, the Conclusions wording has been toned down, This section (lines 665-673) now reads:

“Conclusions

Our study highlights inflammatory changes in mRNA and protein levels that distinguish hypoxic and normoxic HCoV-OC43-infected lung epithelial cells, and identifies possible regulatory mechanisms for further investigation. Our work also identifies two potential therapeutic targets, VEGF and CCL20, which are synergistically-upregulated by hypoxia during HCoV-OC43 infection, targeting of which, for example by modulating hypoxia signaling pathways such as the HIF-1 pathway which regulates both these proteins, may be of utility in controlling hyperinflammation and inhibiting cytokine storm in patients with severe COVID-19.”

  1. The term “synergistic regulation” is used throughout. Authors should define the exact statistical threshold more clearly upfront and consistently refer to it.

The details covering this definition can be found in methods section 2.4 starting from line 149.

Genes were classified as synergistically regulated if the fold change in the HOC43 condition exceeded twice the sum of the fold changes observed in the individual conditions”.

And again, in the results section starting from line 279.

“We defined gene expression changes as synergistic when the fold change of a gene in the HOC43 condition was at least twice the sum of fold changes seen in the HAlone and NOC43 conditions, normalized to NAlone”.

In addition, to take into account the reviewer’s point, an additional reminder (underlined below) of the definition has been added to the first paragraph of the relevant section near the start of the discussion, in lines 443-444. The relevant section now reads:

“Synergistic upregulation.

Importantly, the total number of differentially expressed genes (DEGs) was greater in the HOC43 condition (4551 genes) than the sum of the genes regulated by either stressor alone (HAlone and NOC43) (4252 genes, Figure 1C), suggesting synergistic regulation (in which the fold change caused by hypoxia and virus together is more than double the sum of the fold changes caused by either factor alone) of a number of genes, involving hypoxia and virus infection combined.”

  1. Emphasize limitations of using HCoV-OC43 in translational relevance.

A paragraph justifying the use of HCoV-OC43 and highlighting the limitations in this area due to the differences between the pathogenicity and tropism of this virus and SarsCoV2 has been added to the discussion addressing these points, starting from line 630-647.

“The use of HCoV-OC43 in this study is grounded in practical and mechanistic considerations. HCoV-OC43 shares several features with SARS-CoV-2, including aspects of genomic organization, replication, and host immune modulation, making it a suitable surrogate for studying conserved coronavirus-host interactions. HCoV-OC43 typically causes mild upper respiratory illness and does not typically cause the severe pulmonary pathology associated with early strains of SARS-CoV-2. However, more recently identified SARS-CoV-2 variants, such as Omicron and its sublineages, have demonstrated a shift in tropism toward the upper respiratory tract, aligning more closely with HCoV-OC43 infection. This convergence strengthens the use of HCoV-OC43 as a comparative model, but care should be taken as to not extrapolate findings without further validation.  While HCoV-OC43 is a virus with lower pathogenicity compared to SARS-CoV-2, molecular epidemiologic evidence indicates that this virus arose via a zoonotic transmission event, possible moving into the human population from cattle, around 1890. It has been speculated, although not proven, that this virus could have been responsible for the “Russian Flu” pandemic which occurred in 1889-1890 and caused at least 1 million excess deaths worldwide. The current low pathogenicity of the now-endemic HCoV-OC43 virus may also in part be due to exposure earlier in life, as children can respond very differently to viruses compared to adults, as seen for COVID-19.”

  1. Consider additional validation in primary cells or organoid models.

A valid point, and discussion of the desirability of validating the data in in vivo models, primary cells and organoids has been added in the discussion section, starting from line 658-661.

To complement and expand the in vitro data presented here, extension of the work into animal models, primary cell cultures and organoid models could further enhance our understanding of how hypoxia modulates inflammatory responses to HCoV-OC43 infection.

  1. Provide experimental support for IRES hypothesis or tone down conclusions.

This section has been toned down and made more speculative, and more of an interesting target for future work, starting from line 614-618.

Thus, the increased expression of certain proinflammatory cytokines during coronavirus infections in hypoxia could be, at least in part, dependent on the presence of an IRES within the 5’UTR of the mRNA, and experimental investigation of this interesting possibility which our data highlight represents an exciting future avenue of investigation.”

  1. Streamline figures and reorganize supplementary data for clarity.

We thank the reviewer for noting that there are areas for improvement in the data presentation and are liaising with the journal staff on this point to identify specific changes which could be made to improve clarity and presentation as part of the manuscript finalization process in coordination with the publisher.

Round 2

Reviewer 2 Report

Comments and Suggestions for Authors

I congratulate the authors for doing the commendable job and addressing all my queries.